# Antibacterial and Physicochemical Properties of Orthodontic Resin Cement Containing ZnO-Loaded Halloysite Nanotubes

**DOI:** 10.3390/polym15092045

**Published:** 2023-04-25

**Authors:** Jeong-Hye Seo, Kwang-Mahn Kim, Jae-Sung Kwon

**Affiliations:** 1Department and Research Institute of Dental Biomaterials and Bioengineering, Yonsei University College of Dentistry, Seoul 03722, Republic of Korea; jhseo318@yuhs.ac (J.-H.S.); kmkim@yuhs.ac (K.-M.K.); 2BK21 PLUS Project, Yonsei University College of Dentistry, Seoul 03722, Republic of Korea

**Keywords:** halloysite nanotubes, ZnO, orthodontic resin cement, antimicrobial activity, *S. mutans*

## Abstract

Demineralized white lesions are a common problem when using orthodontic resin cement, which can be prevented with the addition of antibacterial substances. However, the addition of antibacterial substances such as zinc oxide alone may result in the deterioration of the resin cement’s functions. Halloysite nanotubes (HNTs) are known to be biocompatible without adversely affecting the mechanical properties of the material while having the ability to load different substances. The purpose of this study was to prepare orthodontic resin cement containing HNT fillers loaded with ZnO (ZnO/HNTs) and to investigate its mechanical, physical, chemical, and antibacterial properties. A group without filler was used as a control. Three groups containing 5 wt.% of HNTs, ZnO, and ZnO/HNTs were prepared. TEM and EDS measurements were carried out to confirm the morphological structure of the HNTs and the successful loading of ZnO onto the HNTs. The mechanical, physical, chemical, and antibacterial properties of the prepared orthodontic resin cement were considered. The ZnO group had high flexural strength and water absorption but a low depth of cure (*p* < 0.05). The ZnO/HNTs group showed the highest shear bond strength and film thickness (*p* < 0.05). In the antibacterial test, the ZnO/HNTs group resulted in a significant decrease in the biofilm’s metabolic activity compared to the other groups (*p* < 0.05). ZnO/HNTs did not affect cell viability. In addition, ZnO was cytotoxic at a concentration of 100% in the extract. The nanocomposite developed in this study exhibited antimicrobial activity against *S. mutans* while maintaining the mechanical, physical, and chemical properties of orthodontic resin cement. Therefore, it has the potential to be used as an orthodontic resin cement that can prevent DWLs.

## 1. Introduction

Orthodontic treatment is a commonly performed dental procedure that often uses brackets and wires. During such treatment, managing the biofilm that forms around the fixed orthodontic brackets is essential. On the negative management side, due to the interaction between bacteria and the biofilm, demineralization progresses around the fixed orthodontic device that is semi-permanently attached to the tooth during the treatment period, which can lead to an increase in demineralized white lesions (DWLs) and caries [1]. DWLs are areas of tooth enamel that appear opaque due to the loss of minerals and dissolve minerals to form microscopic pores on the tooth surface, unlike the surrounding healthy tooth structure. It is caused by a long-term biofilm around the bracket [2,3]. Previous studies have shown that the incidence of DWLs accounts for more than 60% of orthodontic patients, which can result in increased tooth decay [4].

Orthodontic resin cement provides physical and chemical bonding between the teeth and brackets. Previous studies have added antibacterial substances as fillers in order to effectively suppress the formation of biofilms and consequent DWLs. Recently, research strategies providing antibacterial properties by adding various metal nanoparticles to orthodontic resin cements have been considered [5,6]. Metal nanoparticles such as zinc oxide, silver, titanium dioxide, calcium phosphate, and cerium oxide have an antibacterial effect, penetrating the cell membrane of attached bacteria and killing them. Among metal oxides with antibacterial activity, zinc oxide nanoparticles are human-friendly, nontoxic, and have a high antibacterial effect due to their large specific surface area [7]. The antibacterial activity of ZnO causes membrane disruption by transporting reactive oxygen species resulting from the electrostatic interaction of zinc ions with the biofilm surface [8]. In addition, Zn^2+^ ions interfere with the action of enzymes, inhibiting sugar metabolism [9]. However, the opacity of zinc oxide particles can have a negative effect on photopolymerization, and as nanoparticles also have high cohesion, it results in a reduction in their specific surface area and, consequently, a reduction in their antibacterial ability. Therefore, a solution to compensate for this is needed [10,11].

Recently, various studies have been conducted on halloysite, a mineral nanotube [12]. Halloysite (Al_2_Si_2_O_5_(OH)_4_·2H_2_O) (also known as a halloysite nanotube, HNT) is a tubular structure made of aluminosilicate. As a natural mineral, it is readily available and biocompatible because it is abundant in supply (i.e., thousands of tons) and widely distributed in the earth’s crust, such as in Australia, China, and Switzerland [13,14]. In addition, since the HNTs of the aspect ratio of the surface area are large, it has an elastic modulus and tensile strength that can withstand high loads and is a potential material with high utilization as a nanocomposite [15,16,17]. Functional improvement can be obtained by depositing mediator molecules onto the modified surface of HNTs, and as a natural nanocarrier, continuous release and effective drug delivery are possible [18]. However, studies on adding nanocomposites with antibacterial power to orthodontic resin cement are incomplete.

When comparing the compositions of commercially available light-curing orthodontic resin cement in previous studies, the commercial resin cement containing the matrix of Bis-GMA and TEGDMA had excellent mechanical strength [19]. Based on this, we intend to produce orthodontic resin cement with a filler in which ZnO is deposited on HNTs to see the antibacterial effect and to confirm the clinically required mechanical, physical, and chemical properties of resin cement. The null hypotheses are as follows: (1) there will be no difference in the antibacterial properties of orthodontic resin cement with ZnO and ZnO/HNTs filler; (2) there will be no difference in mechanical, physical, and chemical properties between orthodontic resin cement with ZnO and ZnO/HNTs filler.

## 2. Materials and Methods

### 2.1. Materials

Bisphenol A glycerolate dimethacrylate (Bis-GMA), tri (ethylene glycol) dimethacrylate (TEGDMA), camphorquinone (CQ), ethyl-4-dimethylaminobenzoate (4-EDMAB), zinc oxide nanopowder (<100 nm particle size), and urea were purchased from Sigma-Aldrich Co. (St. Louis, MO, USA). Zinc nitrate hexahydrate was purchased from Duksan Chemicals Co. (Ansan, Republic of Korea). Halloysite nanotubes (HNTs) were provided by Saejeon Int’L Inc. (Seoul, Republic of Korea). All materials were used without further purification.

### 2.2. Synthesis of ZnO/HNTs

Halloysite nanotubes (HNTs) were washed with ethanol and dried at 60 °C for 2 h before use. The fabrication of the ZnO/HNTs nanocomposite was carried out as follows: HNTs 2.4 g, zinc nitrate hexahydrate 3.2 g, and urea 3.2 g were ultrasonically dispersed in 50 mL of distilled water using an ultrasonic device (SH-100; Saehan Ultrasonic, Seoul, Republic of Korea) for 15 min, stirred at 500 rpm for 3.5 h at 95 °C, and calcined at 400 °C for 4 h [12].

### 2.3. Characterization of ZnO/HNT Filler

The ZnO/HNTs structural analysis was investigated using transmission electron microscopy (TEM) (JEM-F200, JEOL, Tokyo, Japan) to confirm the deposition, size, and distribution of ZnO on the HNTs. Prior to TEM analysis, ZnO/HNTs filler ultrasonically dispersed in distilled water at a mass fraction of ~1% was dropped onto carbon-coated copper grids using a micropipette and allowed to dry for 1 day before use [20,21]. Energy dispersive X-ray spectroscopy (EDS) (JEM-F200, JEOL, Tokyo, Japan) was used for elemental analysis of ZnO/HNTs at an accelerating voltage of 200 kV.

### 2.4. Resin Manufacturing

Bis-GMA/TEGDMA (69.5/29.5 wt.%) and CQ/4-EDMAB (0.5/0.5 wt.%) photoinitiators were used as the resin matrix. The filler ratio for use in the orthodontic resin cement was determined to be 5 wt.% according to the results of previous studies [22]. A group consisting of only the resin matrix was set as a control, and 5 wt.% each of HNTs, ZnO, and ZnO/HNTs was added as a filler to the prepared resin matrix. The detailed resin matrix and filler ratios are shown in Table 1 and Table 2.

Bis-GMA, TEGDMA, CQ, and 4-EDMAB were stirred at 1000 rpm for 5 h at 40 °C and then stored at room temperature for 1 day until air bubbles had dispersed. Each HNTs, ZnO, and ZnO/HNTs filler was mixed and added to the prepared resin substrate using a speed mixer (Hauschild SpeedMixer DAC 150, FlackTech Inc., Hauschild, Germany) at 3500 rpm for 1 min. When manufacturing the resin, the experiment was conducted in a dark room, and it was used without delay after mixing and curing the matrix and filler (Figure 1).

### 2.5. Streptococcus mutans Culture

*Streptococcus mutans* (ATCC 25175, *S. mutans*) was subcultured on Brain Heart Infusion Broth (BHI) (Difco, Le Point de Claix, France) medium in a 37 °C incubator. Absorbance was measured using a microplate spectrophotometer (Epoch; BioTek, Winooski, VT, USA) at 600 nm with an OD value of 0.5. The suspension was inoculated into specimens sterilized with EO gas. One milliliter of the suspension was cultured in each well for 14 days on six disk-shaped specimens with a diameter of 10 mm and a height of 2 mm in a 24-well plate (SPL Life Science, Pocheon, Republic of Korea). The medium was replaced every two days.

### 2.6. S. mutans Biofilms of Specimens

Before taking SEM images, bacteria were cultured on the specimen for 1 day and then fixed overnight at 4 °C using 1 mL of 2% glutaraldehyde (Sigma-Aldrich, St. Louis, MO, USA). For dehydration of the specimen, it was immersed in 1 mL of 10, 25, 50, 75, and 90% ethanol for 20 min, respectively, and treated in 100% ethanol for 1 h. After drying completely, a platinum coating (CRESSINGTON SPUTTER COATER 108 auto, Cressington Scientific Instruments Ltd., Oxhey, UK) was applied at 30 mA for 100 s, and SEM (JSM-IT500HR, JEOL Ltd., Tokyo, Japan) was analyzed.

### 2.7. Antibacterial Test

To evaluate the adhesion ability of *S. mutans* to orthodontic resin cement specimens, MTT [3-(4,5-dimetylthiazol-2-yl)-2,5-dyphenyltetrazolium bromide] analysis using metabolic activity was performed. Specimens cultured with *S. mutans* for 1, 3, 5, 7, and 14 days were carefully washed twice in 1 mL of phosphate-buffered saline 1X (PBS) (Welgene, Gyeongsan, Republic of Korea) to remove unattached organisms. Bacteria attached to the specimen were removed using ultrasonic waves in 1 mL of fresh PBS for 10 min. Then, 100 μL was transferred to a 96-well plate, and 10 μL of MTT (1 mg/mL) (Sigma-Aldrich, St. Louis, MO, USA) solution was added. After blocking the light with tinfoil, specimens were stored in a 37 °C incubator for 3 h. To dissolve MTT [3-(4,5-dimetylthiazol-2-yl)-2,5-dyphenyltetrazolium bromide] formazan, 100 μL of dimethyl sulfoxide (DMSO) (Sigma-Aldrich, St. Louis, MO, USA) was added to each well and stored at room temperature for 30 min. The absorbance was measured and analyzed at 570 nm using a microplate spectrophotometer. The values of absorbance were recorded after subtracting blanks.

### 2.8. Cell Cytotoxicity

*L929* (ATCC, CCL-1) was cultured in minimum essential medium (MEM) (Thermo Scientific, Waltham, MA, USA) containing 10% fetal bovine serum (FBS) (Thermo Scientific, Waltham, MA, USA) and 1% penicillin/streptomycin (Gibco, Grand Island, NY, USA) under 5% CO_2_ and 100% humidity in a 37 °C incubator. When the cells reached 80% confluency, they were separated with trypsin-EDTA (Welgene, Daegu, Republic of Korea), seeded in a 96-well plate at a density of 1 × 10^4^ cells/well, and cultured for 24 h.

Disk-shaped specimens with a diameter of 10 mm and a thickness of 2 mm were sterilized with EO gas and extracted for 24 h in a 37 °C incubator under the condition of 3 cm^2^/mL according to ISO 10993-12:2012.

The cytotoxicity test was conducted according to ISO 10993-5:2009. The old medium was carefully removed, and 100 μL per well was administered at 25%, 50%, 75%, and 100% extraction according to the concentration. After 24 h, 10 μL of WST solution (EZ-Cytox Cell Viability Assay Kit, DAEILLAB, Seoul, Republic of Korea) was added, and absorbance was measured at 450 nm after 30 min. It was calculated using the following formula:(1)$$Cellviability\left(\%\right)=\frac{\left({OD}_{exp}-{OD}_{blank}\right)}{\left({OD}_{nc}-{OD}_{blank}\right)}\times 100$$
where OD*_exp_* is and OD*_nc_* is the optical density of the negative control group.

### 2.9. Zn^2+^ Ion Release Concentration

Disk-shaped specimens with a diameter of 10 mm and a thickness of 2 mm were stored in groups of three in a shaking incubator at 37 °C for 1, 3, 5, 7, and 14 days, respectively, according to ISO 10993-12:2012. It was analyzed using an inductively coupled plasma mass spectrometer (ICP-MS) (NexION2000, PerkinElmer Inc., Shelton, CT, USA) to measure the ion release concentration of Zn^2+^.

### 2.10. Surface Roughness

The surface roughness and 3D images (256 × 256 pixels resolution) of disk-shaped specimens with a diameter of 10 mm and a thickness of 2 mm were measured by atomic force microscopy (AFM) (NX-10, ParkSystems, Suwon, Republic of Korea). The measurement was conducted in non-contact mode (NCM) using an NCHR cantilever (ParkSystems, Suwon, Republic of Korea) with a resonance frequency of 3000 kHz. Images were obtained through the XEI program (ParkSystems, Suwon, Republic of Korea).

### 2.11. Depth of Cure

The depth of cure test was conducted according to ISO 4049:2019. The material used in the experiment was inserted into a stainless steel mold with a length of 6 mm and a diameter of 4 mm, and the top of the mold was covered with a transparent film and slide glass and irradiated vertically for 20 s using a light irradiator (Elipar™ DeepCure-L Curing Light, 3M ESPE, Seefeld, Germany). After that, the uncured part was immediately scraped off with a plastic spatula. Six specimens per group were measured for length using a micrometer with an accuracy of 0.1 mm, and the value was divided by two.

### 2.12. Film Thickness

Two glass plates with a contact surface area of 200 mm^2^ were used, and their thickness was measured using a digital caliper (Isomaster 0-25, TESA, Renens, Switzerland) at micrometer (μm) accuracy. A small amount of resin cement was placed between the glass plates, and a load of 150 N was applied vertically with a load device. After 180 s, the load was removed, and the thickness of the two glass plates was measured. The film thickness was calculated as the difference between the two values. Six specimens per group were used, and the procedure was conducted according to ISO 4049:2019.

### 2.13. Water Absorption and Solubility

Water absorption and solubility were tested according to ISO 4049:2019. Six specimens per group were prepared using discoid stainless-steel molds with a diameter of 15 mm and a length of 1 mm. A light irradiator was used to irradiate for 20 s each in nine overlapping areas. After 22 h in a 37 °C desiccator and 2 h in a 23 °C desiccator, this was repeated until a constant mass was obtained. After drying at 37 °C for 22 h and at 23 °C for 2 h, measurements were repeated until the same mass (m1) was obtained with an accuracy of 0.01 mg. The diameter and thickness of the specimen were measured with an accuracy of 0.01 mm, and the volume (V) was calculated. The dried specimens were stored in distilled water at 37 °C for 7 days and weighed (m_2_). The final weight (m_3_) was measured by drying until a constant mass was obtained again. The water absorbance (*W_sp_*) and solubility (*W_sl_*) ratios were calculated using the following equations, respectively:(2)$$W_{sp}=\frac{m_{2}{-m}_{3}}{V}$$
(3)$$W_{sl}=\frac{m_{1}{-m}_{3}}{V}$$

### 2.14. Shear Bond Strength

Ten bovine teeth per group were fixed in self-curing acrylic resin blocks. The tooth surface was polished in the order of 300, 600, 1000, and 1200 grit silicon carbide (SiC) paper, treated with 37% phosphoric acid (Scotchbond™ Universal Etchant, 3M ESPE, Seefeld, Germany) for 60 s, washed with water for 30 s, and dried with air flow for 30 s [1,19]. Primer (Transbond™ XT primer, 3M Unitek, Monrovia, CA, USA) was applied to the tooth surface by lightly blowing air, and then the bracket (Micro-arch Appliances Formula-R; Roth Type, Tomy Intl Inc., Tokyo, Japan) was attached to the tooth with a load of 450.14 g using adhesive (Transbond™ XT Adhesive, 3M Unitek, Monrovia, CA, USA) in the center of the tooth. Excessive adhesive around the bracket was removed, and light was irradiated for 20 s each in the distal and mesial directions. Teeth were stored in distilled water at 37 °C for one day. The shear bond strength was measured using a universal testing machine (Instron 5942, Instron, Norwood, MA, USA). The edge of the blade rod and the tooth-to-bracket adhesive were positioned parallel to each other. A load was applied using a 1 kN load cell at a crosshead speed of 1.0 mm/min, and the result was recorded as N (newton) and converted to MPa (megapascal) as the surface area of the bracket (9.1 mm^2^). The following formula was applied: MPa = N/(9.1 mm^2^)

### 2.15. Flexural Strength

To test flexural strength, specimens were fabricated using a stainless-steel mold of 25 mm × 2 mm × 2 mm. After covering the top and bottom of the specimens with transparent film and slide glass, they were irradiated for 20 s using a light irradiator so that half of the diameter was overlapping. After removing the mold, they were stored in distilled water at 37 °C for 1 day. All specimens were polished using 600-grit silicon carbide (SiC). Testing was conducted according to ISO 4049:2019 using a universal testing machine (Instron 5942, Instron, Norwood, MA, USA) with a crosshead speed of 1.0 mm/min and a 1 kN load cell with 10 specimens per group. Flexural strength (*FS*) was calculated using the following equation:(4)$$FS=\frac{3Fl}{{2wh}^{2}}$$
where *F* is the maximum fracture load (N), *l* is the distance between supports (20 mm), *w* is the width of the specimen (mm), and *h* is the height of the specimen (mm).

### 2.16. Statistical Analysis

Statistical analysis was performed using SPSS Statistics 23 software (IBM, Armonk, NY, USA). One-way ANOVA and Tukey’s post hoc analysis were performed for surface roughness, depth of cure, shear bond strength, flexural strength, and water absorption, and Welch’s ANOVA and Dunnett’s T3 test were performed for film thickness, water solubility, and antibacterial tests.

## 3. Results

### 3.1. Characterization of ZnO/HNT Filler

The size, structure, and chemical composition of the filler are displayed in Figure 2. From the TEM image, it was confirmed that ZnO nanoparticles with sizes between 1 and 10 nm were precipitated on the surface of HNTs with a length of 1 μm and an outer diameter of 100 nm. The EDS image confirmed the presence of the elements Si, O, and Al, which are the components of HNTs, and confirmed that the Zn element was uniformly distributed. The ZnO/HNTs filler was prepared with a weight ratio of HNTs:ZnO of 3:4. As a result of the elemental spectrum, the amount of Zn deposited on the HNTs was 7.28 wt.% and the atomic amount was 2.25 wt.%.

### 3.2. Antibacterial Ability

Figure 3 is the result of analyzing the metabolic activity of resin cement using *S. mutans*. Depending on the period, the ZnO and ZnO/HNTs groups showed statistically significant differences compared to the control group (*p* < 0.05). It was confirmed that the ZnO/HNTs group showed the highest antibacterial activity by presenting lower biofilm metabolic activity than the ZnO group. The metabolic activity of *S. mutans* according to the period indicated a statistical difference from the 3rd day in the ZnO/HNTs group and from the 5th day in the ZnO group. In addition, there was no difference in biofilm metabolic activity between the 7th and 14th days in the ZnO group, but there was a difference in the ZnO/HNTs group (*p* < 0.05).

### 3.3. S. mutans Biofilms of Specimens

Figure 4 is an SEM image confirmed by 5000× magnification of the bacteria attached to the specimen. In all specimens, it was confirmed that the spherical *S. mutans* was extensively attached. *S. mutans* from groups other than the ZnO/HNTs group had a chain length of more than 5 μm, and the ZnO/HNTs group showed a chain length of less than 5 μm and a relatively small number of bacteria.

### 3.4. Cell Cytotoxicity

Figure 5 shows the cell viability according to the concentration of the extract for 24 h. A negative control, a positive control, and a medium-only blank were used for group comparison. *L929* cell viability was not affected at extract concentrations of 25%, 50%, and 75% in all groups. However, when 100% extract was injected, the ZnO group showed relatively low cell viability, and the ZnO/HNTs group showed high cell viability.

### 3.5. Zn^2+^ Ion Release Concentration

Figure 6 shows the concentration of Zn^2+^ ion release by group according to the period. The ZnO group released ions rapidly compared to the ZnO/HNTs group on the first 1 and 3 days, but the amount of Zn^2+^ ions released after 3 days gradually decreased. However, in the ZnO/HNTs group, the amount of Zn^2+^ ions released continuously increased for 3 days.

### 3.6. Surface Roughness

Table 3 and Figure 7 show the roughness of each group of specimens. Surface roughness (Ra) through AFM analysis did not show a significant difference between groups (*p* > 0.05). The 3D surface of the specimen could be visually confirmed through the AFM image. Compared to the ZnO and ZnO/HNTs groups, the control and HNT groups showed relatively flat and even surface images.

### 3.7. Chemical Properties

As shown in Figure 8a, the depth of cure was found to be lower in the group containing ZnO. There was no significant difference between the control and HNTs groups, and there was a significant difference between the ZnO and ZnO/HNTs groups. However, the ZnO/HNTs group (2.196 ± 0.037) showed a significantly higher polymerization depth than the ZnO group (1.178 ± 0.035) (*p* < 0.05).

### 3.8. Physical Properties

The film thickness exhibited a statistically significant difference in all groups, and the group containing HNTs indicated higher results. As shown in Figure 8b, the lowest film thickness of the control group, without filler addition (1.3 ± 0.5), and the highest film thickness of the ZnO/HNTs group (68.0 ± 5.8) were indicated (*p* < 0.05).

Table 4 presents water absorption and solubility. The ZnO group showed the highest water absorption, with the ZnO/HNTs group showing the lowest absorption (*p* < 0.05). The group containing HNTs showed lower water absorption. There was no significant difference between groups in terms of solubility (*p* > 0.05).

### 3.9. Mechanical Properties

The results of the mechanical properties of shear bond strength and flexural strength are shown in Figure 9. For the shear bond strength, the existing material and the experimental group were compared by adding the fillers to orthodontic resin cement used in clinical applications. The shear bond strength of the group containing ZnO/HNTs (29.206 ± 2.210) was the highest, and there was a statistically significant difference compared to the control group (24.874 ± 2.765), which is an existing product (*p* < 0.05). In flexural strength, the control and ZnO groups exhibited statistically higher strength values compared to the HNTs and ZnO/HNTs groups. However, the ZnO/HNTs group (91.511 ± 13.949) indicated relatively higher flexural strength than the HNTs group (86.662 ± 10.334) (*p* < 0.05).

## 4. Discussion

Depending on whether orthodontic brackets are adhered or not, the incidence of microorganisms is affected. Orthodontic patients have an increased risk of developing dental lesions compared to before orthodontic treatment [23]. In response to this, it is necessary to develop a material that can inhibit the growth of microorganisms by imparting antibacterial properties to orthodontic resin cement.

Therefore, this study aimed to suppress the metabolic activity of bacteria and prevent biofilm formation by adding a nanocomposite with antibacterial properties and to develop a material that satisfies the mechanical, physical, and chemical properties corresponding to orthodontic resin cement. To date, researchers have made extensive attempts to synthesize dental materials with antibacterial properties, but studies on whether their antibacterial effects persist are lacking [24,25]. Therefore, this study additionally tested the antibacterial sustainability of orthodontic resin cement.

HNTs have been proven to be biocompatible materials with high potential for use as fillers, and many studies have been conducted to load them with antibacterial substances [26,27]. According to previous studies, as a result of experiments with 0–10 wt.% of filler added to the resin, the group with 5 wt.% added demonstrated the most ideal antibacterial and mechanical properties. As the filler content increased, the antibacterial activity increased, but the mechanical properties decreased, showing the opposite result [20]. Accordingly, this study conducted experiments by setting the filler content to 5 wt.% for all groups except the control group. The antibacterial activity of ZnO causes membrane disruption by transporting reactive oxygen species resulting from the electrostatic interaction of zinc ions with the biofilm surface [8]. *S. mutans* is a Gram-positive bacterium that causes initial carious lesions by attaching to the tooth’s surface. In this study, the biofilm metabolic activity of attached *S. mutans* was analyzed using MTT, which was performed to confirm the growth and survival of microorganisms against antimicrobial substances [28]. The ZnO/HNTs group revealed a significant difference in the inhibition of microbial attachment compared to the other groups. In addition, since the time of initial attachment is important for biofilm formation, the adhesion ability of bacteria on the first day of attachment was observed through SEM images [29]. Through this, the results of the quantitative antimicrobial test for each group were qualitatively and visually confirmed. As a result, the least amount of s.mutan was seen in the ZnO/HNTs group. The first null hypothesis was rejected because the biofilm metabolic activity of *S. mutans* according to the attachment period was lower in the ZnO/HNTs group than in the ZnO group. In addition, the reason for the decrease in bacteria in the control group on day 14 was hypothesized as follows. The first may be due to the toxicity of bis-GMA used as a resin matrix. Previous studies have shown that bis-GMA has the highest cytotoxicity [30]. The adhesive strength may have been low because the surface roughness of the second resin cement was low. Finally, it may be due to the existence of bacterial growth limitations in a controlled experimental environment. In the experiment, the same mass fraction of filler was added to the orthodontic resin cement, but the pure ZnO content of the ZnO and ZnO/HNTs groups could be different. However, even though a small amount of ZnO was loaded onto the HNTs, the antibacterial effect was higher than that of the ZnO group. This demonstrated that the HNTs influenced the outcome by maximizing the action of the mediator, inducing sustained release [31]. To prove this, an analysis of the release concentration of Zn^2+^ ions during the same period as the antibacterial test was performed. Compared to the ZnO group, the continuous ion release ability of the ZnO/HNTs group could be quantitatively confirmed. ZnO nanoparticles release a significant amount of Zn^2+^ ions at once, which can slightly reduce their antibacterial activity as particle concentration increases due to high cohesive force [32]. In the ZnO/HNT group, the content of ZnO particles was lower than that of the case where only ZnO particles were added by depositing ZnO on the HNT, but the release amount was continuously increased, supporting the antibacterial results.

Surface roughness can have important effects on bacterial adhesion and biofilm formation [33]. As a result of the experiment, it was confirmed that there was no significant difference between the groups. It can be interpreted that the surface roughness of the specimen did not affect the difference between groups.

The cell viability, that is, the degree of toxicity induction, can be confirmed through the components of the resin cement and the extract of the added material [34]. In addition, materials with high antibacterial activity require evaluation of cell viability. Excluding the ZnO group with a 100% extract concentration, it showed a value of 70% or higher according to ISO 10993-5:2009. The ZnO group added at 5 wt.% in 100% extract was toxic. In contrast, the ZnO/HNTs group deposited ZnO and did not induce cell toxicity because the content was relatively low. This demonstrated that the ZnO/HNTs group did not affect cell growth by depositing ZnO using biocompatible HNTs. The depth of cure can be affected by the scattering and absorption of light due to the photoinitiator, refractive index, and color of the material [35]. All groups resulted in a polymerization depth of 1.5 mm or greater according to ISO 4049:2019. The ZnO group showed significant differences from all the other groups. This may have resulted in a relatively low depth of cure due to the high turbidity of the ZnO particles themselves. However, the HNTs group was not significantly different from the resin-cement matrix control group. This is considered to have had a positive effect on the depth of cure because the HNTs were transparent and had high light conductivity.

The particles, content, size, and shape of the filler can be considered factors that affect film thickness [36]. A thicker film thickness of the adhesive can affect the decrease in bonding strength, and previous studies have shown that the thinner the film thickness, the less load is removed during bracket debonding [37,38]. According to the TEM image analysis results, the particle sizes of ZnO and HNTs are 1–10 nm and 1 μm, respectively, indicating that HNTs are relatively larger than ZnO. Accordingly, it is expected that the HNTs group would possess the highest film thickness. In addition, there was no significant difference from the film thickness average (68.3 ± 2.8 μm) of the commercial resin cement used in the study; thus, the ZnO/HNTs group is suitable for use as an orthodontic resin cement.

Water absorption and solubility vary depending on the type and chemical structure of the resin matrix, the content or area of the filler, and the binding force with the matrix [39]. High water absorption and solubility may result in low strength or stress, and the absorbed water may interfere with the bonding with the filler. In addition, it is necessary to test water absorption and solubility because dissolved substances can cause toxicity in vivo [40]. In this study, all group results showed lower water absorption and solubility. In terms of water absorption, the ZnO/HNTs group presented relatively low results. This can be explained by the large specific surface area of the resin cement filler compared to other groups.

The mechanical strength test is necessary to check the ability to withstand the occlusal force or external load applied to the part where the tooth enamel and the bracket are joined. In this study, flexural and shear bond strengths were tested to confirm the mechanical properties. The factors that affect flexural strength are related to the content of the filler and the bond with the matrix [41]. The flexural strength of all experimental groups showed a high strength of over 50 MPa (average 113.91 ± 29.47 MPa), which is the ISO 4049:2019 standard. Before testing the shear bond strength, we tried to observe the usability of the filler by adding each experimental filler to commercially available orthodontic resin cement. The resin cement used in the experiment had a filler content of 70–80 wt.%, whereas the resin cement produced in this study only had a filler content of 5 wt.% [42]. Additionally, the film thickness may vary, potentially affecting debonding results. To address this issue, we supplemented our experiments with the addition of fillers produced from commercial resin cement. Etching and primer were used when attaching the orthodontic bracket to create the same environment as the clinical technique. Effective orthodontic bracket adhesion is 7 MPa or more [43]. As a result of this study, all groups were over 20 MPa (average 29.47 ± 4.66 MPa), and an important finding was that the shear bond strength of the ZnO/HNTs group was the highest. Previous research has shown that the larger the size of the filler, such as a hybrid or nanocluster, the higher the bonding force and ability to resist fracture by transforming accumulated stress [44,45]. When ZnO/HNTs filler was added to commercially available resin cement, a significant difference was shown from the control group, which can be interpreted as a positive effect of ZnO/HNTs as a filler. Therefore, the second null hypothesis was rejected because there were differences in mechanical, physical, and chemical properties between the ZnO and ZnO/HNTs groups.

## 5. Conclusions

In this study, ZnO/HNTs filler was added to orthodontic resin cement to impart antibacterial properties. In addition, we aimed to develop a material that could maintain mechanical, physical, and chemical properties. When ZnO/HNTs were added, the antibacterial activity was sustained, and excellent mechanical strength was shown. In conclusion, ZnO/HNTs filler can be used as an ideal biocompatible filler for orthodontic resin cement to prevent caries.

## Figures and Tables

**Figure 1 polymers-15-02045-f001:**
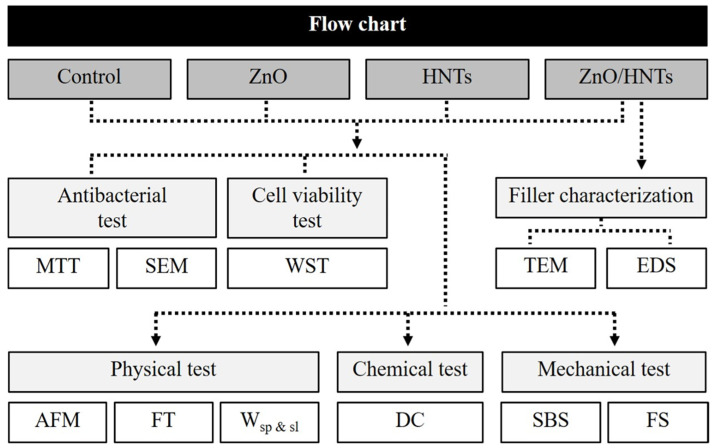
The flow chart of the overall experiment.

**Figure 2 polymers-15-02045-f002:**
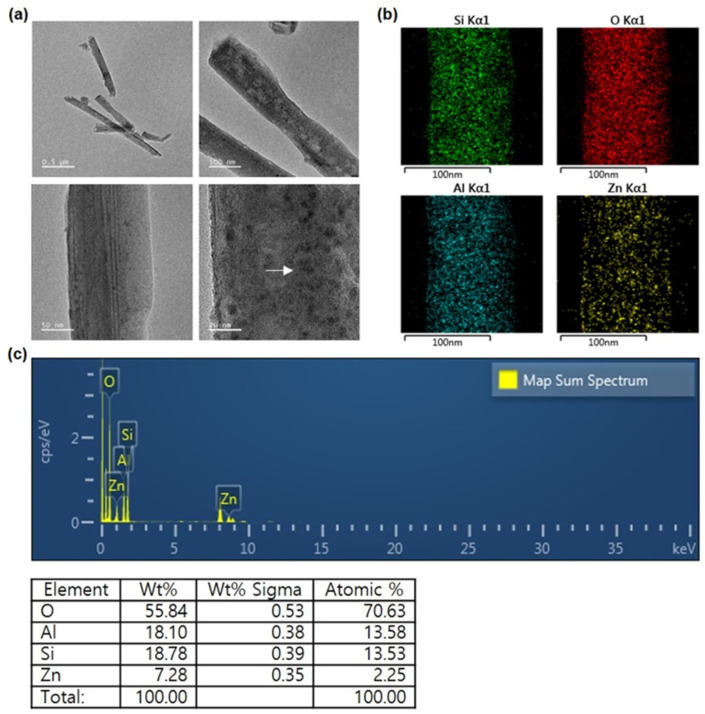
(**a**) TEM images of ZnO/HNTs at various magnifications; (**b**) EDS mapping of Si, O, Al, and Zn elements; and (**c**) elemental spectrum and content.

**Figure 3 polymers-15-02045-f003:**
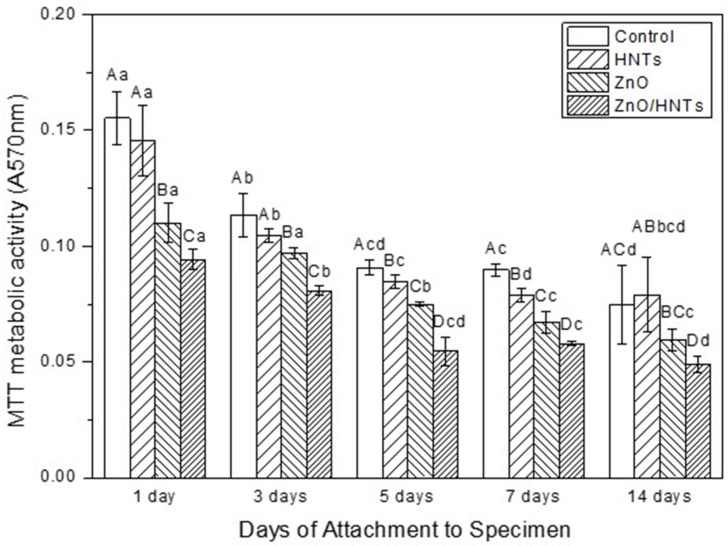
MTT metabolic activity. Other capital letters indicate statistical differences between groups (*p* < 0.05). Other lowercase letters indicate statistical differences across time periods (*p* < 0.05).

**Figure 4 polymers-15-02045-f004:**
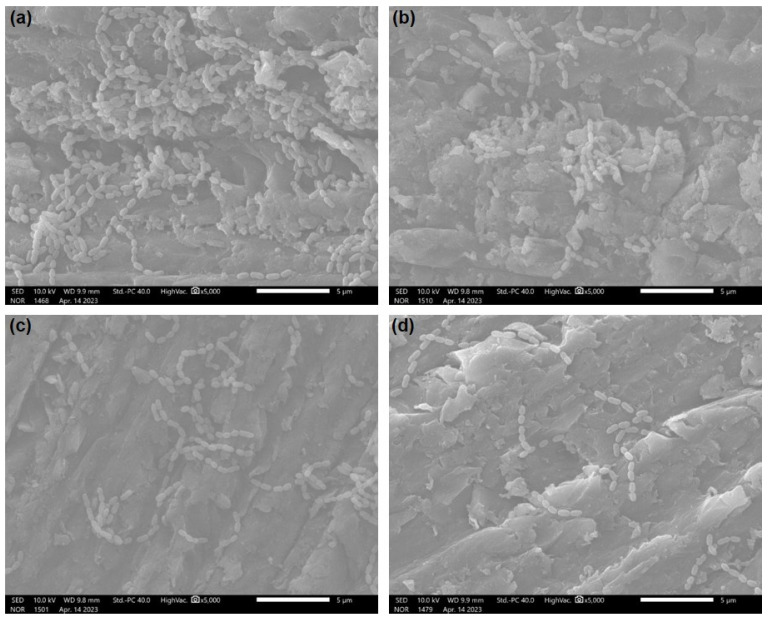
SEM images: (**a**) control group; (**b**) HNTs group; (**c**) ZnO group; and (**d**) ZnO/HNTs group (5000× magnification).

**Figure 5 polymers-15-02045-f005:**
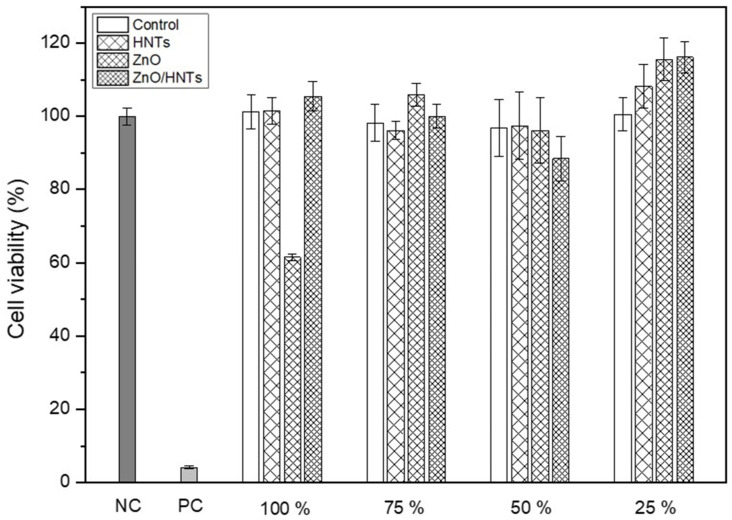
Cell viability of experimental groups.

**Figure 6 polymers-15-02045-f006:**
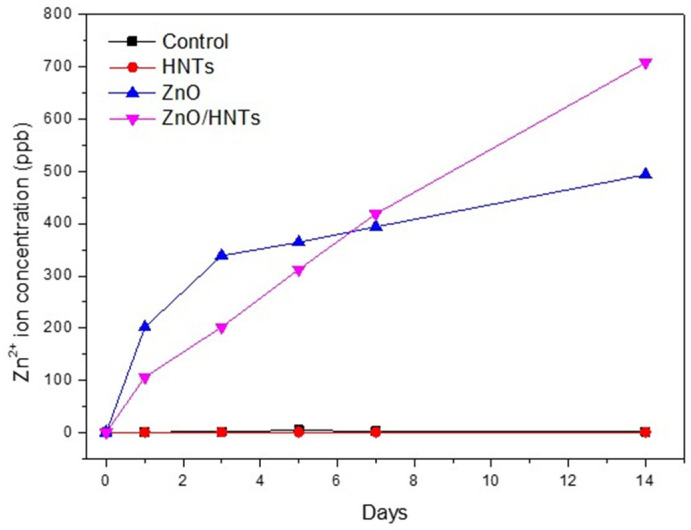
Zn^2+^ ion release concentration according to period.

**Figure 7 polymers-15-02045-f007:**
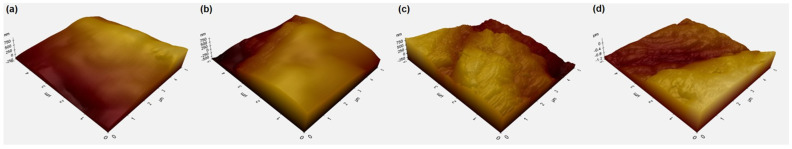
AFM images: (**a**) control group; (**b**) HNTs group; (**c**) ZnO group; and (**d**) ZnO/HNTs group.

**Figure 8 polymers-15-02045-f008:**
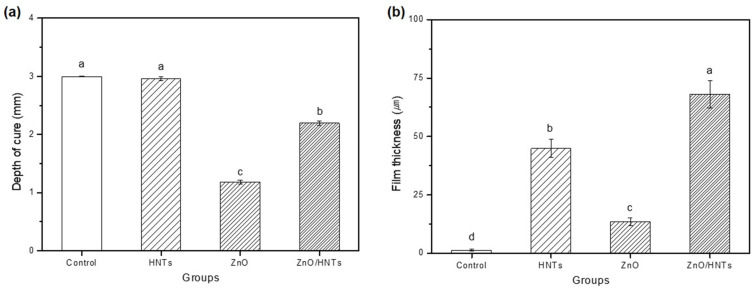
(**a**) Depth of cure; (**b**) film thickness. Other lowercase letters indicate statistical differences between groups (*p* < 0.05).

**Figure 9 polymers-15-02045-f009:**
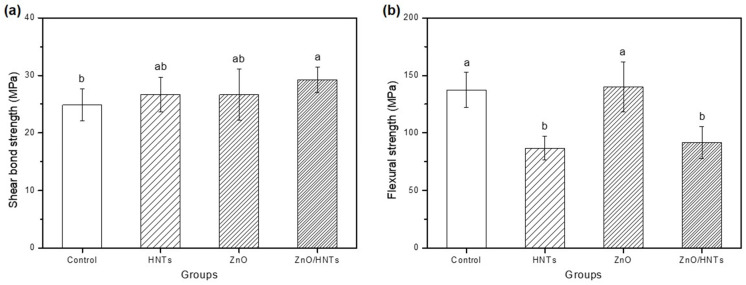
(**a**) Shear bond strength; (**b**) flexural strength. Other lowercase letters indicate statistical differences between groups (*p* < 0.05).

**Table 1 polymers-15-02045-t001:** Resin matrix composition.

Resin Matrix	Bis-GMA	TEGDMA	CQ	4-EDMAB
Composition (wt.%)	69.5	29.5	0.5	0.5

**Table 2 polymers-15-02045-t002:** Resin cement composition.

Group Code	Resin Matrix (wt.%)	Filler (wt.%)
Control	100	0
HNTs	95	5
ZnO	95	5
ZnO/HNTs	95	5

**Table 3 polymers-15-02045-t003:** Average surface roughness.

Group Code	Control	HNTs	ZnO	ZnO/HNTs
Ra (μm)	0.232 ± 0.106 ^a^	0.209 ± 0.059 ^a^	0.239 ± 0.106 ^a^	0.260 ± 0.133 ^a^

^a^ Other lowercase letters indicate statistical differences between groups (*p* < 0.05).

**Table 4 polymers-15-02045-t004:** Water sorption and solubility.

Group Code	Water Sorption (μg/mm^3^)	Water Solubility (μg/mm^3^)
Control	9.893 ± 0.353 ^ab^	0.587 ± 0.203
HNTs	9.430 ± 0.279 ^bc^	0.496 ± 0.227
ZnO	10.027 ± 0.261 ^a^	0.329 ± 0.176
ZnO/HNTs	9.175 ± 0.335 ^c^	0.480 ± 0.120

^abc^ Other lowercase letters indicate statistical differences between groups (*p* < 0.05).

## Data Availability

Not applicable.

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
