# Peer review of "Antibacterial and Physicochemical Properties of Orthodontic Resin Cement Containing ZnO-Loaded Halloysite Nanotubes"

_polymers, 2023, doi:10.3390/polym15092045_

Round 1

Reviewer 1 Report

In this study, the authors aim to suppress the metabolic activity of bacteria and inhibit biofilm formation by adding a nanocomposite with antibacterial properties. They also aim to develop a robust orthodontic resin cement material that satisfies mechanical, physical, and chemical properties. The study is well conducted, but it is not suitable for publication in its current form. Please attend to the following observations:

  1. The authors include SEM-EDS characterization of the material. The EDS is not quantitative. Please indicate the amount of ZnO deposited in the HNTs. 
  2. Please include an AFM characterization of the different films.
  3. Antibacterial activity: a) include images of bacterial growth in the different films; b) why on day 14, the number of bacteria decreases considerably in the control group (compared to treated groups).
  4. Evaluate ion lixiviation. After 14 days, resin cement is still active (antibacterial activity). Is it suitable for practical applications?
  5. The authors claim the biocompatibility of ZnO-HNTs but do not demonstrate such a statement. Please include a biocompatibility study. 

Author Response

Reviewer 1

Q1

The authors include SEM-EDS characterization of the material. The EDS is not quantitative. Please indicate the amount of ZnO deposited in the HNTs.

A1

Thank you for your valuable comments. Based on your comments, the content of Zn has been added to the results.

The ZnO/HNTs filler was prepared with a weight ratio of HNTs:ZnO of 3:4. As a result of the elemental spectrum, the amount of Zn deposited on the HNTs was 7.28 wt.% and the atomic amount was 2.25 wt.%.

Q2

Please include an AFM characterization of the different films.

A2

Thank you for your good suggestion. Based on your suggestion, AFM characterization was performed, and the following contents were added to the thesis.

2.10. Surface roughness

The surface roughness and 3D images (256 × 256 pixels resolution) of disk-shaped specimens with a diameter of 10 mm and a thickness of 2 mm were measured by atomic force microscopy (AFM) (NX-10, ParkSystems, Suwon, Korea). The measurement was conducted in non-contact mode (NCM) using an NCHR cantilever (ParkSystems, Suwon, Korea) with a resonance frequency of 3000 kHz. Images were obtained through the XEI program (ParkSystems, Suwon, Korea).

3.6. Surface roughness

Table 3 and Figure 7 show the roughness of each group of specimens. Surface roughness (Ra) through AFM analysis did not show a significant difference between groups (p > 0.05). The 3D surface of the specimen could be visually confirmed through the AFM image. Compared to the ZnO and ZnO/HNTs groups, the Control and HNT groups showed relatively flat and even surface images.

Table 3. Average surface roughness.

Group Code

Control

HNTs

ZnO

ZnO/HNTs

Ra (μm)

0.232 ± 0.106a

0.209 ± 0.059a

0.239 ± 0.106a

0.260 ± 0.133a

1 Other lowercase letters indicate statistical differences between groups (p < 0.05).

Figure 7. AFM images (a) Control group; (b) HNTs group; (c) ZnO group; (d) ZnO/HNTs group.

The surface roughness can have important effects on bacterial adhesion and biofilm formation [33]. As a result of the experiment, it was confirmed that there was no significant difference between the groups. It can be interpreted that the variable of surface roughness of the specimen did not affect the difference between groups.

Q3

Antibacterial activity: a) include images of bacterial growth in the different films; b) why on day 14, the number of bacteria decreases considerably in the control group (compared to treated groups).

A3

Thank you for your comments. According to your comments, a) the SEM images of the surface of the specimens with bacteria were analyzed, and b) the reduction of bacteria in the control group on the 14th day was explained.

a) 2.6. S. mutans biofilms of specimens

Before taking SEM images, bacteria were cultured on the specimen for 1 day and then fixed overnight at 4 °C using 1 mL of 2% glutaraldehyde (Sigma-Aldrich, St. Louis, MO, USA). For dehydration of the specimen, it was immersed in 1 mL of 10, 25, 50, 75, and 90% ethanol for 20 minutes, respectively, and treated in 100% ethanol for 1 hour. After drying completely, a platinum coating (CRESSINGTON SPUTTER COATER 108 auto, Cressington Scientific Instruments Ltd., Oxhey, England) was applied at 30 mA for 100 seconds, and SEM (JSM-IT500HR, JEOL Ltd., Tokyo, Japan) was analyzed.

3.3. S. mutans biofilms of specimens

Figure 4 is an SEM image confirmed by 5000 X magnification of the bacteria attached to the specimen. In all specimens, it was confirmed that spherical S. mutans was extensively attached. S. mutans of groups other than the ZnO/HNTs group had a chain length of more than 5 μm, and the ZnO/HNTs group showed a chain of less than 5 μm and a relatively small number of bacteria.

Figure 4. SEM images (a) Control group; (b) HNTs group; (c) ZnO group; (d) ZnO/HNTs group (5000 X magnification).

In addition, since the time of initial attachment is important for biofilm formation, the adhesion ability of bacteria on the first day of attachment was observed through SEM images [29]. Through this, the results of the quantitative antimicrobial test of each group were qualitatively and visually confirmed. As a result, the least amount of s.mutan was seen in the ZnO/HNTs group.

b) In addition, the reason for the decrease in bacteria in the control group on day 14 was hypothesized as follows. The first may be due to the toxicity of Bis-GMA used as a resin matrix. Previous studies have shown that Bis-GMA has the highest cytotoxicity [30]. The adhesive strength may have been low because the surface roughness of the second resin cement was low. Finally, it may be due to the existence of bacterial growth limitations in a controlled experimental environment.

Q4

Evaluate ion lixiviation. After 14 days, resin cement is still active (antibacterial activity). Is it suitable for practical applications?

The authors claim the biocompatibility of ZnO-HNTs but do not demonstrate such a statement.

A4

Thank you for your good suggestion. According to your suggestion, we confirmed the release of Zn ions for 14 days. We have added content to the paper as below.

2.9. Zn2+ ion release concentration

Disk-shaped specimens with a diameter of 10 mm and a thickness of 2 mm were stored in groups of three in a shaking incubator at 37 °C for 1, 3, 5, 7, and 14 days, respectively, according to ISO 10993-12:2012. It was analyzed using inductively coupled plasma mass spectrometer (ICP-MS) (NexION2000, PerkinElmer Inc., Shelton, USA) to measure the ion release concentration of Zn2+.

3.5. Zn2+ ion release concentration

Figure 6 shows the concentration of Zn2+ ion release by group according to the period. The ZnO group released rapidly compared to the ZnO/HNTs group on the first 1 and 3 days, but the amount of Zn2+ ions released after 3 days gradually decreased. However, in the ZnO/HNTs group, the amount of Zn2+ ions released continuously increased from the 3 days.

Figure 6. Zn2+ ion release concentration according to period.

To prove this, an analysis of the release concentration of Zn2+ ion during the same period as the antibacterial test was performed. Compared to the ZnO group, the continuous ion release ability of the ZnO/HNTs group could be quantitatively confirmed. ZnO nanoparticles release a significant amount of Zn2+ ions at once, which can slightly reduce their antibacterial activity as particle concentration increases due to high cohesive force [32]. In the ZnO/HNT group, the content of ZnO particles was lower than that of the case where only ZnO particles were added by depositing ZnO on the HNT, but the release amount was continuously increased, supporting the antibacterial results.

Q5

Please include a biocompatibility study.

A5

Thank you for your sincere advice. Following your advice, a cytotoxicity test was conducted and the contents are as follows.

2.8. Cell cytotoxicity

L929 (ATCC, CCL-1) was cultured in minimum essential medium (MEM) (Thermo Scientific, Waltham, MA, USA) containing 10 % fetal bovine serum (FBS) (Thermo Scientific, Waltham, MA, USA) and 1 % penicillin/streptomycin (gibco, Grand Island, NY, USA) under 5 % CO2 and 100% humidity in a 37 °C incubator. When the cells reached 80 % confluency, they were separated with Trypsin-EDTA (Welgene, Daegu, Korea), seeded in a 96-well plate at a density of 1 × 104 cells/well, and cultured for 24 hours.

Disk-shaped specimens with a diameter of 10 mm and a thickness of 2 mm were sterilized with EO gas and extracted for 24 hours in a 37 °C incubator under the condition of 3 cm2/mL according to ISO 10993-12:2012.

The cytotoxicity test was conducted according to ISO 10993-5:2009. The old medium was carefully removed, and 100 μL per well was administered at 25%, 50%, 75%, and 100% extraction according to the concentration. After 24 hours, 10 μL of WST solution (EZ-Cytox Cell Viability Assay Kit, DAEILLAB, Seoul, Korea) was added, and absorbance was measured at 450 nm after 30 minutes. It was calculated using the formula.

3.4. Cell cytotoxicity

Figure 5 shows the cell viability according to the concentration of the extract for 24 hours. A negative control, a positive control, and a medium-only blank were used for group comparison. L929 cell viability was not affected at extract concentrations of 25%, 50%, and 75% in all groups. However, when 100% extract was injected, the ZnO group showed relatively low cell viability, and the ZnO/HNTs group showed high cell viability.

Figure 5. Cell viability of experimental groups.

The cell viability, that is, the degree of toxicity induction, can be confirmed through the components of the resin cement and the extract of the added material [34]. In addition, materials with high antibacterial activity require evaluation of cell viability. Excluding the ZnO group with 100% extract concentration, it showed a value of 70% or higher according to ISO 10993-5:2009. The ZnO group added at 5 wt.% in 100% extract was toxic. In contrast, the ZnO/HNTs group deposited ZnO and did not induce cell toxicity because the content was relatively low. This demonstrated that the ZnO/HNTs group did not affect cell growth by depositing ZnO using biocompatible HNTs.

Reviewer 2 Report

The manuscript talks about somewhat interesting topic, but definitely needs to be improved before being ready for publication. In some sections the flow of experiment is not similar to others, for e.g., the discussion section is not coherent with the result section in some places. Below are a few more points which can help improve the quality of the paper:

Abstract:

-Spell out WSL- Line 26.

Introduction:

-Please improve the definition of DWL.

- In line 42, clarify that not all metal nanoparticles have antibacterial properties.

-Explain the mechanism by which ZnO nanoparticles exhibit antibacterial properties. Move the explanation from discussion to introduction. 

- Where can halloysite be easily found?

- Why this particular resin was used?

Methods: 

-In section 2.1, add in details about materials used for preparing nanoparticles and doing cell culture.

-Add details on how EDS was done.

-After adding the filler material to the resin, how long was the mixture left to dry/ set?

- Why weren't the cured specimens used for antibacterial testing?

-In line 107, "length" of 2mm but be replaced by "height" of 2mm.  

-What is the purpose of conducting experiment 2.8?

-Line 145-146: an accuracy of 0.01mg... needs to be written.

-Why a different resin was used in the mechanical strength testing? Better to use same material for all the testing. 

-Please provide more details on the bracket that was used.

-Add some pictures/ schematics to make it easier for the readers to understand the experimental setup better.

Results:

- The figures numbers are wrong in the text as compared to the figure labels. Must be corrected.

- The use of letters to differentiate between significantly different groups must be explained more clearly.

- Figure 1, there is no proof of nanoparticles. From EDS it looks like all the zinc is uniformly distributed on the nanotubes. Use arrows to point to specific nanoparticles in a zoomed in figure of nanotubes.

- For testing antibacterial properties, it is important for the material to have antibacterial properties in the initial implantation time points as well. comment on day 1 antibacterial activity as well. 

-Did the resin themselves have any antibacterial activity? why did the metabolic bacterial activity of control groups decrease at day 14?

Author Response

Reviewer 2

Q1

Abstract:

-Spell out WSL- Line 26.

Introduction:

-Please improve the definition of DWL.

A1

Thank you for your careful check and review. Sorry for the confusion. We have modified the content as below.

- Therefore, it has the potential to be used as an orthodontic resin cement that can prevent DWLs.

-DWLs are areas of tooth enamel that appear opaque due to loss of minerals and dissolve minerals to form microscopic pores on the tooth surface, unlike the surrounding healthy tooth structure. It is caused by a long-term biofilm around the bracket [2,3].

Q2

Introduction:

- In line 42, clarify that not all metal nanoparticles have antibacterial properties.

-Explain the mechanism by which ZnO nanoparticles exhibit antibacterial properties. Move the explanation from discussion to introduction.

A2

Sorry for the confusion in writing the manuscript. Not all metal nanoparticles are antibacterial. Edited and added to the introduction.

The antibacterial activity of ZnO causes membrane disruption by transporting reactive oxygen species resulting from the electrostatic interaction of zinc ions with the biofilm surface [8]. In addition, Zn2+ ions interfere with the action of enzymes, inhibiting sugar metabolism [9].

Q3

Introduction:

- Where can halloysite be easily found?

A3

Thank you for your comments. Sorry for not conveying information about HNTs. We modified the content as follows.

As a natural mineral, it is readily available and biocompatible because it is abundant in supply, thousands of tons, and widely distributed in the earth's crust, such as Australia, China, and Switzerland [13,14]. In addition, since the HNTs of the aspect ratio of the surface area is large, it has the elastic modulus and tensile strength that can withstand high loads and is a potential material with high utilization as a nanocomposite [15-17].

Q4

Introduction:

- Why this particular resin was used?

A4

Thank you for your helpful comments. Based on your comments, the content has been reflected in the introduction. Added details as follows.

When comparing the compositions of commercially available light-curing orthodontic resin cement in previous studies, the commercial resin cement containing the matrix of Bis-GMA and TEGDMA, had excellent mechanical strength [19].

Q5

Methods:

-In section 2.1, add in details about materials used for preparing nanoparticles and doing cell culture.

A5

Thank you for your valuable input. We modified materials for nanoparticle fabrication in 2.1, and materials used for bacteria and cells are detailed in 2.5, 2.6, 2.7, and 2.8. Thanks again for your kind advice.

Q6

Methods:

-Add details on how EDS was done.

A6

Thank you for your detailed advice. The lack of information provided by EDS is as follows.

Energy dispersive X-ray spectroscopy (EDS) (JEM-F200, JEOL, Tokyo, Japan) was used for elemental analysis of ZnO/HNTs at an accelerating voltage of 200 kV.

Q7

Methods:

-After adding the filler material to the resin, how long was the mixture left to dry/ set?

- Why weren't the cured specimens used for antibacterial testing?

A7

Thank you for your comments. The production process of orthodontic resin cement and the use of specimens are as follows.

-When manufacturing the resin, the experiment was conducted in a dark room, and it was used without delay after mixing and curing the matrix and filler.

-After hardening the specimen for the antibacterial test, EO gas sterilization was performed to prevent external variables.

Q8

Methods:

-In line 107, "length" of 2mm but be replaced by "height" of 2mm.

-Line 145-146: an accuracy of 0.01mg... needs to be written.

A8

Sorry for the mistake of using inappropriate words. Missing content has been corrected. Thank you for your careful review.

-One milliliter of the suspension was cultured in each well for 14 days on 6 disk-shaped specimens with a diameter of 10 mm and a height of 2 mm in a 24-well plate (SPL Life Science, Pocheon, Korea).

- After drying at 37 °C for 22 hours and at 23 °C for 2 hours, measurements were repeated until the same mass (m1) was obtained with an accuracy of 0.01 mg. The diameter and thickness of the specimen were measured with an accuracy of 0.01 mm and the volume (V) was calculated.

Q9

Methods:

-What is the purpose of conducting experiment 2.8?

A9

Thank you for your valuable comments. We have written an explanation in the discussion based on your comments.

A thicker film thickness of the adhesive can affect the decrease in bonding strength, and previous studies have shown that the thinner the film thickness, the less the load removed during bracket debonding [37,38].

Q10

Methods:

-Why a different resin was used in the mechanical strength testing? Better to use same material for all the testing.

A10

Thank you for your comments. We have made the following corrections to the discussion based on your comments. Thanks again for your advice.

The resin cement used in the experiment had a filler content of 70-80 wt.%, whereas the resin cement produced in this study only had a filler content of 5 wt.% [42]. Additionally, the film thickness may vary, potentially affecting debonding results. To address this issue, we supplemented our experiments with the addition of fillers produced from commercial resin cement.

Q11

Methods:

-Please provide more details on the bracket that was used.

A11

Thank you for your valuable comments. Bracket information is as follows.

~,and then the bracket (Micro-arch APPLIANCES Formula-R; Roth Type, Tomy Intl Inc, Tokyo, Japan) was attached~

Q12

Methods:

-Add some pictures/ schematics to make it easier for the readers to understand the experimental setup better.

A12

Thank you for your kind review of the comments. According to your recommendation, we have created a flow chart to help understand the study.

Figure 1. The Overall flow chart of the experiment.

Q13

Results:

- The figures numbers are wrong in the text as compared to the figure labels. Must be corrected.

- The use of letters to differentiate between significantly different groups must be explained more clearly.

A13

Sorry for the confusion with our mistake in not matching the figure number label in the figure number and text. We added letters indicating significant differences in water solubility. And we modified the sentence to say "Other lowercase letters indicate statistical differences between groups (p<0.05)." Once again thank you for your detailed review.

Q14

Results:

- Figure 1, there is no proof of nanoparticles. From EDS it looks like all the zinc is uniformly distributed on the nanotubes. Use arrows to point to specific nanoparticles in a zoomed in figure of nanotubes.

A14

Thank you for your kind and detailed comments. Following your advice, We changed the nanoparticles in Figure 1(a) to a verifiable clear image, and inserted an arrow.

Figure 2. (a) TEM images of ZnO/HNTs at various magnifications; (b) EDS mapping of Si, O, Al, Zn elements; (c) Elemental Spectrum and Content.

Q15

Results:

- For testing antibacterial properties, it is important for the material to have antibacterial properties in the initial implantation time points as well. comment on day 1 antibacterial activity as well.

A15

Thank you for your valuable comments. According to your comments, we identified specimens adhered for 1 day through SEM analysis. Its contents are as follows.

In addition, since the time of initial attachment is important for biofilm formation, the adhesion ability of bacteria on the first day of attachment was observed through SEM images [29].

Q16

Results:

-Did the resin themselves have any antibacterial activity? why did the metabolic bacterial activity of control groups decrease at day 14?

A16

Thank you for your comments. Based on your comments, we further explained the bacterial reduction in the control group on day 14 in the discussion as follows.

In addition, the reason for the decrease in bacteria in the control group on day 14 was hypothesized as follows. The first may be due to the toxicity of Bis-GMA used as a resin matrix. Previous studies have shown that Bis-GMA has the highest cytotoxicity [30]. The adhesive strength may have been low because the surface roughness of the second resin cement was low. Finally, it may be due to the existence of bacterial growth limitations in a controlled experimental environment.

Round 2

Reviewer 1 Report

The authors properly attend reviewer´s comments. The manuscript is ready to be published